# The Interrelationship between Diabetes Mellitus and Emotional Well-Being: Current Concepts and Future Prospects

**DOI:** 10.3390/healthcare12141457

**Published:** 2024-07-22

**Authors:** Polyxeni Mangoulia, Charalampos Milionis, Eugenia Vlachou, Ioannis Ilias

**Affiliations:** 1Faculty of Nursing, National and Kapodistrian University of Athens, GR-11527 Athens, Greece; pmango@nurs.uoa.gr; 2Department of Endocrinology, Diabetes, and Metabolism, Elena Venizelou General Hospital, GR-11521 Athens, Greece; pesscharis@hotmail.com; 3Department of Nursing, University of West Attica, GR-12243 Athens, Greece; evlachou@uniwa.gr; 4Department of Endocrinology, Hippokration General Hospital, GR-11527 Athens, Greece

**Keywords:** diabetes mellitus, biopsychosocial models, mental disorders, psychosocial functioning, comorbidity, self-care

## Abstract

Diabetes mellitus is a lifelong metabolic disorder that impacts people’s well-being and biopsychosocial status. Psychiatric problems and diabetes mellitus have a complex, reciprocal interaction in which one condition affects the other. In this narrative review, we provide an overview of the literature on the psychological effects of diabetes, expound on the evaluation of emotional disorders in the setting of diabetes, and suggest interventions aimed at enhancing both mental and physical health. Diabetes can make daily life complicated and stressful. Frequent blood glucose testing, taking medications on a regular basis, adhering to a tight diet plan, and exercising are some examples of the suggested daily routine of subjects with diabetes. Furthermore, comorbid diseases and typical diabetic complications can have a detrimental impact on quality of life. When mental health conditions coexist with diabetes mellitus, there is a greater likelihood of medication noncompliance, a decreased commitment to diabetes-related self-care, increased functional impairment, inadequate glycemic control, a higher risk of complications, and overall higher healthcare expenses. Thus, evaluation of the mental health status of patients with diabetes is crucial. When treating psychological issues and psychiatric disorders, a comprehensive biopsychosocial approach should be taken, and where appropriate, psychopharmacological therapies or psychotherapy should be applied. The goal of continuous education and assistance for self-care is to give individuals with the disease the information and abilities they need to control their condition over time.

## 1. Introduction

Psychological stress is common in many physical illnesses and has been increasingly identified as a risk factor for the onset and progression of diabetes mellitus. The latter is a chronic metabolic disorder that drastically affects the biopsychological identity and well-being of individuals. Therefore, the management of diabetes should extend far beyond simply treating hyperglycemia. The support of patients with diabetes should aim to assist them in acquiring the necessary knowledge and skills to diachronically meet the demands of their disease, including coping with the relevant emotional burden and potential mental pathologies.

### 1.1. Classification, Pathophysiology, and Epidemiology of Diabetes

Diabetes mellitus consists of a range of metabolic disorders that present hyperglycemia as their principal characteristic. The different types of this clinical syndrome are pathogenically related to the interaction of environmental and genetic factors. Diabetes mellitus falls into two primary categories, according to the pathophysiological mechanisms that underlie the development of hyperglycemia. Type 1 diabetes mellitus is caused by absolute insulin insufficiency resulting from autoimmunity against components of pancreatic beta cells. Type 2 diabetes mellitus is characterized by insulin resistance, decreased insulin secretion, and elevated hepatic glucose synthesis at varying levels. Other clinical entities of diabetes mellitus result from genetic disorders in insulin secretion or activity, anomalies hindering insulin secretion, impairments to the mitochondria, and circumstances compromising glucose tolerance [1].

Chronic hyperglycemia is linked to potentially serious secondary complications in several tissues and organs. Diabetic complications are responsible for the majority of morbidity and mortality associated with the disease. Diabetic microvasculopathy manifests abnormal growth and permeability of microcirculatory vessels, including arterioles, capillaries, and venules [2]. It pathologically affects the small blood vessels in the glomeruli, retina, heart, skin, and muscle [3]. Furthermore, diabetes mellitus seems to participate in the etiology of macrovasculopathy by causing functional and structural alterations in major blood vessels [4]. Unfortunately, complications from diabetes can seriously impair a person’s quality of life [5]. Governments, health organizations, and clinicians should advocate for the rights of those with diabetes in order to help them live healthy and productive lives.

The current epidemiological state and future estimated trajectory of diabetes are alarming. Throughout the previous forty years, the prevalence of this disease has increased globally, and this trend is predicted to continue in the years to come. At present, there are over 460 million individuals worldwide who suffer from the disease, and this figure is expected to rise by 25% and 51% in 2030 and 2045, respectively [6]. Type 2 diabetes accounts for the majority (over 85%) of total cases. The rising global trends of obesity and unhealthy lifestyles fuel the relevant prevalence. The diabetes-induced premature morbidity, reduced life expectancy, and financial burden render the disease a significant public health concern [7]. Diabetes can also profoundly burden emotional wellness with a direct impact on social life [8,9].

### 1.2. Diabetes Mellitus and Mental Health

Diabetes mellitus and emotional state can interact in harmful ways. Apparently, there is a bidirectional relationship between the diabetic condition and mental health. Indeed, diabetes mellitus is a chronic metabolic disorder with a potentially serious psychological impact on the affected individuals. The somatic malady of the disease, in combination with the demands and prospects of a life with diabetes, may cause distress. In reverse, mood disorders and psychosocial impairments among patients with diabetes can influence the development and outcomes of hyperglycemia because of the difficulty in maintaining appropriate self-management and health behaviors [10]. Both diabetes and mental/psychological dysphoria can lead to restrictions in the physical and emotional functionality of those who suffer, forming a self-feeding cycle of interaction, as depicted in Figure 1. Intrauterine life, genetics, social determinants, and medication may also play a role in this interaction [11].

Stress is a state of tension and agitation that emerges as the result of an organism’s effort to respond to environmental dangers appropriately. However, prolonged activation of stress mechanisms can lead to serious detrimental effects on the body. In this regard, both physical and psychological stress have been thought to influence the progress of diabetes via behavioral and physiological pathways. Emotional strain is linked to unhealthy lifestyle patterns, such as bad nutritional habits, physical inactivity, smoking, alcohol abuse, and deficient self-care. Unhealthy behaviors are, in turn, associated with a greater risk of exhibiting diabetes and also with poorer glycemic control [12]. In parallel, protracted physiological stress induces chronic activation of the hypothalamic–pituitary–adrenal axis and the sympathetic nervous system. Both the elevated secretion of glucocorticoids and the increased release of catecholamines enhance glucose intolerance. Although acute hyperglycemia during times of stress can benefit the body because it diverts glucose to insulin-independent tissues, chronic stress can lead to insulin resistance and diabetes [13]. Finally, extended periods of stress can trigger or intensify inflammation, which in turn could influence the development of diabetes mellitus through the action of inflammatory cytokines [14]. Figure 2 summarizes the stress-related biological processes that could lead to dysglycemia.

### 1.3. Objectives

The interest in the interrelationship between glycemia and emotional health has grown significantly over the past years. Although there is a bulk of literature regarding the psychological impact of diabetes mellitus, the clinical significance in determining the course of glycemic control is still uncertain. Furthermore, clinicians’ knowledge about the two-way interaction between hyperglycemia and emotional conditions is often scarce. Consequently, the limited understanding of the psychological aspects of diabetes does not permit health professionals to apply interventions in order to improve clinical outcomes and decrease the emotional burden. The present article aims to offer insights into two important fields of inquiry to fill this gap in knowledge. First, it investigates the emotional consequences of living with diabetes, and second, it examines the coexistence of common mental health disorders and hyperglycemia. The ultimate goal is to render clinicians capable of providing a holistic approach to the management of diabetes.

## 2. Methods

A narrative review is a comprehensive and unbounded report on current knowledge about an issue that also contains analysis and interpretation [15,16]. The present work used a narrative approach to summarize the existing evidence on the psychosocial aspects of diabetes, elaborate on the assessment of emotional disorders within the context of diabetes, and propose interventions for the improvement of both physical and mental health. The relevant literature was searched in the Medline and Google Scholar databases in April and May 2024 with the combined use of a series of terms, including “diabetes mellitus”, “distress”, “mental health”, “quality of life”, and “psychological interventions”. The electronic search retrieved numerous titles (3892 emerged from Medline and 1200 were checked from Google Scholar). Among them, articles that served the narrative of this review were purposefully selected and consisted of 78 publications [17,18,19,20,21,22,23,24,25,26,27,28,29,30,31,32,33,34,35,36,37,38,39,40,41,42,43,44,45,46,47,48,49,50,51,52,53,54,55,56,57,58,59,60,61,62,63,64,65,66,67,68,69,70,71,72,73,74,75,76,77,78,79,80,81,82,83,84,85,86,87,88,89,90,91,92,93,94,95,96]. Opinion pieces, case reports, primary research studies, medical guidelines, reviews (both systematic and narrative), and meta-analyses were included (see Table 1). The selected papers either provided useful evidence about certain aspects of the relationship between diabetes and emotional health or recommended strategies for managing mental health issues associated with diabetes.

## 3. Distress from Coping with Diabetes

Living with diabetes is often complex and stressful. The daily routine may include frequent blood glucose measurements, regular medication intake, strict nutritional schedules, and engagement in physical activities. Moreover, common diabetic complications and comorbid conditions can negatively affect the quality of life. In addition to the medical difficulties, the socio-economic demands of treating diabetes can contribute to the emergence of emotional unease. Therefore, it is not surprising that diabetes is associated with the occurrence of “diabetes distress” (see below).

### 3.1. Diabetes-Related Distress

Diabetes is a chronic disease with high demands on patients. Apart from its adverse metabolic consequences, it is associated with serious emotional challenges. Indeed, the life of individuals with diabetes is full of uncertainties and repeated medical, psychological, and social tribulations. Some of these patients may suffer from the so-called “diabetes distress”. The latter refers to the gamut of negative emotions that derive from the burden of living with diabetes [17]. Diabetes distress can be expressed with a series of problematic attitudes and behaviors [18]. The management of diabetes should not focus only on glycemic control and the prevention of medical complications, but also on the relief from the psychological burden. Hence, psychosocial treatment should be incorporated into clinical care and provided to all people with diabetes, with the purpose of achieving the highest attainable health outcome and emotional well-being [19].

Emotional reactions of people with diabetes distress may include fear, worry, rage, guilt, sorrow, and disappointment [20]. However, the symptomatology of diabetes distress can extensively vary in range and intensity between patients. Establishing diagnostic criteria is critical to uniformly identify patients at risk and achieve consistency of findings across studies. There is no widely accepted objective method of defining the existence of diabetes distress. Questionnaires are commonly used for this purpose, with high scores documented to be associated with poor glycemic control, low self-efficacy, unhealthy diet, and limited physical activity [21]. For example, the Diabetes Distress Screening Scale (DDS) is a widely used instrument that yields a total score as well as sub-scale scores in four domains, each addressing a different kind of distress. Other questionnaires are the Problem Areas in Diabetes Scale (PAID), the Well-Being Questionnaire (W-BQ), the Revised Diabetes-Specific Quality of Life Scale (DSQoLs-R), and the Revised Illness Perceptions Questionnaire (IPQ-R) [22]. Judicious use of different scales is required because of the various confounding factors that could influence the occurrence of distress [23].

The prevalence of diabetes distress has been estimated by various studies across countries to far exceed 30% among patients with diabetes [24,25,26,27]. Distinguishing between diabetes distress and depression is often a difficult task. The former is rooted in the everyday reality of living with diabetes, while the latter is a generic sensation of emotional malaise that is unrelated to any particular condition or event [28]. The differential diagnosis requires a thorough history taking, with special attention to the chronological progression of symptoms, their association with psychosocial factors, any previous psychiatric disorder, the onset of hyperglycemia, and the course of glycemic control. Next, the clinical consultation must determine the patient’s perception of diabetes and his/her attitude toward its management [29]. Both diabetes distress and depression increase a patient’s risk of having a poorly managed disease, developing diabetic complications, and experiencing a low quality of life. Therefore, routine psychological assessments are necessary to achieve optimal treatment and prevent fatal consequences [30,31].

### 3.2. Emotional Expressions of Patients with Diabetes

Living with diabetes may generate various expressions of distress by the affected individuals. Unfortunately, the manifestations of diabetes distress are not always identified or adequately addressed by health professionals. Issues that are particularly stressful in coping with diabetes include accepting the diagnosis, managing daily diabetes-related tasks, worrying about complications, and interacting in routine social situations.

#### 3.2.1. Negative Reactions to the Initial Diagnosis

Initially, individuals may encounter difficulties accepting the diagnosis of a life-long condition and possibly respond with anxiety, sadness, anger, defiance, or denial [32]. When diagnosed, a person with diabetes becomes abruptly aware of the fragility of human health. The diagnosis defines a critical point in time that separates the previous “normal” life from the current vulnerable state. Not everyone will react to the diagnosis with the same kind or intensity of feelings. Furthermore, this initial stage varies in duration between individuals. In fact, many people can oscillate back and forth between negative emotions and acceptance, whereas others may never truly recover from the initial shock [33]. A properly knowledgeable clinician can identify the unvoiced feelings of persons with a recent diagnosis of diabetes. Empathetic responses to patient’s reactions allow a more effective therapeutic relationship to evolve [34].

#### 3.2.2. Reduced Adherence to Treatment

Patients with diabetes usually rely on regular intake of medication, adjustments in lifestyle, and frequent visits to medical settings in order to achieve satisfactory glycemic control. Unfortunately, unending needs for both clinical care and self-management can detrimentally affect the mood as well as the quality of life of persons with diabetes [35]. Burnout may even occur when an individual is frustrated due to the tasks of ongoing self-treatment [36,37]. Psychological pressure can, in turn, impede the patient’s ability and willingness to comply with the recommended care [38,39]. This situation may compromise the health status of persons with diabetes. Further psychological concerns may include the reluctance to initiate insulin or other injectable therapy because of an exacerbated phobia of injection pain or due to misconceptions that associate the administration of insulin with treatment failure [40]. With the appropriate support within the clinical environment, changes in mindset can assist in alleviating diabetes-related exhaustion and misbeliefs. In addition to appropriate medical anti-diabetic treatment, addressing the psychological barriers to effective self-control can potentially reduce the somatic consequences for individuals with diabetes [41].

#### 3.2.3. Excessive Fear of Complications

Acute diabetic complications include ketoacidosis (mainly in patients with type 1 diabetes) and a non-ketotic hyperosmolar state (mostly in individuals with type 2 diabetes) [42]. Moreover, the overdose of some anti-diabetic medications (insulin and insulin secretagogue drugs) can induce hypoglycemic episodes [43]. In addition, long-term hyperglycemia can lead to vision impairment, kidney failure, neuropathy, gastrointestinal and genitourinary dysfunction, cardiovascular disease, stroke, foot ulcers, susceptibility to infections, and dermatological manifestations [44]. Persistent worries about complications or deterioration of health status are causative factors of despondency and emotional turmoil. The constant concern about averting diabetic complications either in day-to-day life or in the long term might result in undue anxiety related to avoiding extreme values of blood glucose [45].

#### 3.2.4. Social Exclusion

Social exclusion is a phenomenon in which certain population groups find themselves at a disadvantage that causes them to move away from mainstream society and be unable to participate in normal life [46]. The social activities of patients with diabetes are often compromised. The need to follow scheduled meals, monitor blood sugar levels, and receive regular medication might synergistically erode personal contacts and hamper professional duties. Furthermore, sexual dysfunction may generate disturbed romantic relationships. Diabetes is also associated with a high prevalence of eating disorders [47,48]. In an equitable and inclusive society, addressing the social vulnerabilities of people with diabetes must be a priority for governments, health systems, and clinicians.

## 4. Diabetes and Common Psychiatric Disorders

Diabetes mellitus and psychiatric disorders have a complex relationship, with both influencing each other in multiple ways. The fragmentation of anti-diabetic and psychiatric treatment further complicates the management of both conditions and renders patients susceptible to poor health outcomes. The interplay between diabetes and mental disease is schematically presented in Figure 3.

### 4.1. Psychiatric Comorbidities among Patients with Diabetes

Patients with diabetes may suffer from various mental disorders at an increased rate in comparison with non-diabetic counterparts. Unfortunately, mental health comorbidities of diabetes may hinder blood glucose management and, hence, increase the risk of diabetic complications and impaired quality of life. In addition, the financial cost of care and the social burden of the disease rise. Major depression, anxiety, and eating disorders are the most common psychiatric conditions [49], but further mental problems may disproportionately affect people with diabetes mellitus. From a medical and social point of view, identifying psychiatric comorbidities among patients with diabetes should be a priority because they increase morbidity and mortality and decrease the quality of life.

#### 4.1.1. Depression

Depression is a common and severe mental health disorder, with a lifetime prevalence exceeding 10% [50]. The diagnostic criteria for major depression consist of a core symptom, such as depressed mood or loss of interest or pleasure in daily activities, and a majority of specified symptoms, including feelings of worthlessness or guilt, fatigue, concentration problems, suicidality, abrupt changes in weight or appetite, psychomotor agitation or retardation, and sleeping irregularities, that last for at least two weeks. Depression is one and a half to two times more frequent in people with diabetes compared to the general population [51]. Unfortunately, the outcome of depression and diabetes is worse when they coexist. The simultaneous presence of both conditions is linked to higher odds of diabetic complications [52] and functional disability [53] and results in reduced life expectancy [54].

The connection between diabetes and depression is based on three assumptive connections. First, both diseases might have a common etiology. Abnormal fetal development, socio-economic disadvantage, and unhealthy lifestyles appear to be causes of both conditions through the excessive activation of stress mechanisms. Chronic hypercortisolemia and prolonged sympathetic nervous stimulation can lead to dysglycemia, but they can also provoke physiological and behavioral changes that are involved in depression. Second, diabetes could facilitate the occurrence of depression. Indeed, hyperglycemia causes neurodegenerative changes in the brain, including ischemic encephalopathy and cerebral atrophy. Depression is associated with these processes, especially at the level of the prefrontal cortex and hippocampus. Moreover, diabetes is a state of chronic inflammation that could produce a biological predisposition to depression via effects on synaptic and neuroendocrine function. Apart from these pathophysiological mechanisms, the burden of living with diabetes can be responsible for the presence of depressive symptoms. Third, depression could promote the development of diabetes. In this case, impaired functionality may lead to deleterious living habits, which in turn can increase the risk of diabetes. In addition, the use of some antidepressants, particularly for a long period and in high dosages, may adversely affect blood glucose [55,56,57]. Nonetheless, the existing evidence is preliminary at best, and further research is needed to confirm the underlying causalities.

#### 4.1.2. Anxiety

Anxiety disorders usually involve a persistent and undue feeling of worry or dread, which can interfere with daily life. The prevalence of clinical anxiety among people with diabetes is alarming and reaches 14% [58], whereas it is around 4% in the general population. The link between anxiety and diabetes is not clear, but it is likely due to a synergistic interaction in behavior. On one side, dietary restrictions, complex medication schemes, and continuous monitoring of blood glucose are demanding routines in the everyday lives of patients with diabetes that could foster symptoms of anxiety. On the other side, living with anxiety may negatively influence eating habits and physical activity and predispose one to impaired glucose homeostasis [59]. Although this has not been definitely proven, the coexistence of diabetes and anxiety may be associated with suboptimal glycemic control [60].

#### 4.1.3. Eating Disorders

Eating disorders are characterized by episodes of disordered dietary behavior that occur in the context of underlying disturbances in emotional regulation and body image perception. Extreme expressions may lead to medical complications, including electrolyte abnormalities, arrhythmias, dyslipidemia, endocrine anomalies, dental erosion, and gastrointestinal pathologies. Disturbed eating behavior in individuals with diabetes is of great concern because it can interfere with a patient’s ability to adhere to the daily tasks of self-care and promote hyperglycemia. The constant need for glycemic control is a potential iatrogenic factor that triggers nutritional dysregulation, especially among females with type 1 diabetes [61].

#### 4.1.4. Other Mental Pathologies

Diabetes mellitus often coexists with schizophrenia. The classic risk factors for type 2 diabetes, namely unhealthy nutrition, sedentary lifestyle, and obesity, are frequent among patients with schizophrenia, even at the early stages of the disease. Individuals with schizophrenia frequently have a low socio-economic status, which limits their ability to lead healthy lives. Antipsychotic drugs may also cause hyperglycemia [62]. Hypoglycemia or diabetic ketoacidosis can lead to delirium, in patients with diabetes. The clinical features mainly include decreased psychomotor activity, confusion, and impaired sensory abilities. Hallucinations, sleeping–awakening disorders, and disrupted cognitive function may also be present. The impact of delirium on the health and well-being of individuals with diabetes can be serious because it can prolong hospitalizations, restrict functionality, and increase morbidity and mortality [63]. Finally, it is not certain whether smoking, alcohol overconsumption, and substance use are more common in people with diabetes, but the health effects are particularly burdensome in these cases.

### 4.2. Evaluation of Emotional Health in Patients with Diabetes

The diagnosis of mental disorders requires a detailed history, careful physical examination, and targeted laboratory investigation. Psychiatric symptoms coexisting with diabetes may vary in spectrum, severity, and duration and can precede, occur simultaneously, or follow glycemic dysregulation. In addition, there could be some overlap between the clinical traits of diabetes and the features of mental illnesses. In general, health professionals should be familiar with the emotional, cognitive, and behavioral derangements in patients with diabetes so that they can provide early intervention if needed.

The defining symptoms for each mental illness are detailed in the fifth edition of the Diagnostic and Statistical Manual of Mental Disorders (DSM-5) and alternatively in Chapter V of the tenth revision of the International Statistical Classification of Diseases and Related Health Conditions-10 (ICD-10). Both manuals can serve as guides for clinicians to diagnose mental conditions and for insurance agencies to reimburse care expenses. Scales, such as the Patient Health Questionnaire (PHQ), the Symptom Check-list-90 (SCL-90), and the Hospital Anxiety and Depression Scale (HADS), are sensitive and validated diagnostic tools for common psychiatric disorders like depression and anxiety and can be useful in screening patients with diabetes [64].

## 5. Therapeutic Interventions

The search for better therapeutic outcomes from anti-diabetic interventions is a continuous effort. Comorbidity consisting of diabetes and mood disorders or psychiatric conditions can lead to worse glycemic control, functional impairment, higher mortality, and increased healthcare expenses [65]. Therefore, it is reasonable to assume that mental health support should be an essential part of care for people with diabetes to address potential coexisting emotional impairments and psychological distress. Such interventions are expected to have favorable effects on self-management, the prevention of complications, and life expectancy. Modern care requires the provision of integrated services to people with diabetes by a group of health professionals of different specialties. Of course, coordination and information sharing between the parties involved are important parameters for achieving more effective care. The therapeutic approach for this purpose could be based on two basic tools: psychosocial support and psychopharmacological agents.

### 5.1. Psychosocial Support

Several strategies have been applied to simultaneously improve both physical and mental health in patients with diabetes. Education on self-care, cognitive behavioral therapy, and social endorsement are the most common efforts that have been applied for this purpose. Psychosocial interventions are usually associated with modest improvements in glycemic control and a small relief of anxiety and psychological distress. However, they are significantly more effective in individuals with poor glycemic control and those with very negative emotional states, probably because of the larger leeway for improvement [66]. This is an exceptionally important parameter because psychosocial interventions are ordinarily aimed at people with severe deficits in blood glucose management and/or loss of control of emotions and productive functioning. In these cases, a holistic approach to the treatment of diabetes may assist vulnerable individuals in gaining stronger motivation and better skills for coping with the disease [67,68]. Of course, providing such services requires professionalism and expertise on the part of health staff, as well as funding and arrangements by the health system.

#### 5.1.1. Guidance on Self-Care

The meaning of self-care entails the person’s ability to handle the clinical and psychosocial consequences of living with diabetes mellitus. Thus, educational guidance on self-care should aim to enhance knowledge about somatic complications and mental comorbidities of diabetes and cultivate the skills and motivation needed for optimal glycemic control and emotional adjustment [69]. Self-efficacy in monitoring blood glucose, following an appropriate diet, remaining physically active, adhering to treatment, and avoiding unhealthy behaviors can collectively mitigate the burden of diabetes and, in turn, contribute to a reduction in psychological suffering. For this purpose, properly trained health professionals need to provide individualized instructions that will render each patient self-confident and capable of managing physical and psychosocial challenges [70]. Given that depression is more common following a diagnosis of diabetes, how medical professionals inform patients of their condition and then help those who have recently developed diabetes may have an impact on the patients’ long-term mental health [71]. The implementation of psychoeducational programs for diabetes is therefore necessary. To enhance patients’ awareness of diabetes, clinicians could provide them with oral and written information on disease management, record their progress during visits, and adjust the education accordingly.

#### 5.1.2. Cognitive Behavioral Therapy

The integration of cognitive behavioral therapy into clinical practice puts great emphasis on assessing and enhancing an individual’s understanding of diabetes and its demands for self-management [72]. Cognitive behavioral therapy can assist individuals in reorganizing dysfunctional thoughts and changing harmful behaviors into positive patterns. In this way, it can lead to better mood adjustment [73]. It may be well suited for patients with diabetes mellitus for several reasons. First, it focuses on concepts, such as optimism and resilience, that have been linked to favorable health outcomes. Second, it can be applied to a wide range of people with a variety of emotional problems. Third, it can promote interest in diabetic self-management, which, in turn, may improve self-care and quality of life [74]. Cognitive behavioral therapy should be considered a useful tool for optimizing glycemic control and relieving anxiety and depression in the long term [75].

#### 5.1.3. Establishment of Social Supportive Networks

In a broad sense, social support aims to improve patients’ quality of life by offering advice, information, emotional sustenance, material resources, and aid that strengthen individuals’ abilities to cope with the adversities of their disease. It can be provided by formal bodies, such as insurance agencies, social services, community organizations, and charities, as well as by informal advocates, mainly consisting of family, friends, acquaintances, and patients’ unions. The role of social supportive networks is to facilitate access to care, ensure socio-economic backing, reinforce appropriate health behaviors, and provide practical assistance. In the case of patients with diabetes, the existence of social supportive networks has been associated with positive health outcomes, particularly for individuals experiencing economic deprivation or social isolation [76].

### 5.2. Use of Psychopharmacological Agents

There are no psychiatric treatment algorithms specific to populations with diabetes. Hence, pharmacological management should follow guidelines used for the general population.

#### 5.2.1. Antidepressants

Treatment objectives should concentrate on the remission of depression as well as the improvement of glycemic control. The latter can serve as a predictor for the eventual outcome due to the well-known negative effects of the interaction between depression and diabetes [71]. Psychopharmacological treatment depends on the type, severity, and other characteristics of the psychiatric disorder and is based on the drugs in use, with some basic considerations.

Since depression has an adverse impact on psychological well-being and glycemic outcomes, the treatment of depression in people with diabetes should be directed toward the improvement of both psychological and somatic outcomes [77]. Tricyclic antidepressants (TCAs) have been shown to have negative side effects, including drowsiness, increased appetite, weight gain, and confusion, as well as an increased risk of cardiovascular disease [78,79]. Early research has indicated that weight gain is still a possibility with newer antidepressants, selective serotonin reuptake inhibitors (SSRIs), and serotonin–norepinephrine reuptake inhibitors (SNRIs). However, mirtazapine (an atypical tetracyclic antidepressant) has a lower risk of weight gain than TCAs. Amitriptyline is considered the TCA that causes the most potent weight gain, while paroxetine was recognized to have a higher risk of weight gain among SSRIs [80].

SSRIs, with the exception of paroxetine, are well tolerated by people with diabetes, having a better profile regarding adverse events [77]. More so than SSRIs, SNRIs may impact neuropathic pain. Duloxetine and venlafaxine seem to benefit individuals with diabetes [81,82,83]. Compared to other second-generation antidepressants, bupropion, which has a dopaminergic effect but no serotonergic effect, has a decreased incidence of sexual dysfunction and is not linked to weight gain [84,85,86]. In general, antidepressants are the first line of treatment (SSRIs being the first-choice medication) when treating severe depression, usually in conjunction with psychotherapy. Depending on the needs of each patient, these individuals may receive care as inpatients or outpatients [71].

#### 5.2.2. Antipsychotics

Compared to the general population, those on first- or second-generation antipsychotics have a greater prevalence of diabetes [87]. Research comparing various antipsychotics suggests that the respective risks for diabetes development vary. Patients using olanzapine or clozapine have a consistently greater reported risk of diabetes, while those taking aripiprazole have the lowest risk. Nonetheless, all antipsychotics, with the exception of amisulpride and aripiprazole, are linked to a greater prevalence of diabetes [88]. Weight gain, dyslipidemia, and new-onset diabetes mellitus have been linked to olanzapine and clozapine, and to a lesser extent, quetiapine and risperidone [89,90].

Severe anxiety will require the co-administration of benzodiazepines for symptomatic relief. Their use follows the same rules as in non-diabetic people. Long-acting preparations are superior, especially when there is a need for stable, regular administration. Despite the rarity of benzodiazepine-related clinical metabolic abnormalities, investigation into insulin secretion and action raises the possibility that these medications could eventually increase the risk of glycemic dysregulation [91].

## 6. Recommendations for Clinical Practice

The therapeutic management of diabetes mellitus in modern clinical practice must cease to be a one-dimensional treatment of hyperglycemia. The interdependence of physical and mental health requires the implementation of interventions that will lead to the empowerment of patients with diabetes so that they are able to comply with medical instructions and the recommended lifestyle. This tactic can reduce the possibility of emotional distress, improve the clinical condition of potential psychiatric disorders, lower the risk of diabetic complications, and eventually lead to a better quality of life from both a psychosocial and a physical standpoint.

In this direction, psychoeducation is a psychotherapeutic method with four main activities: information about the disease, training in solving problematic situations, cultivation of communication skills, and enhancement of self-confidence [90]. The goal is to strengthen patients’ resilience to the adversities related to the daily reality of diabetes mellitus. Preliminary evidence is promising, showing that, when conducted properly, psychoeducational interventions successfully reduce diabetes-related distress and hyperglycemia in people with diabetes [92,93]. However, further research is needed to determine the design tailored to the characteristics of each patient, the relative involvement of each kind of specialist, the frequency and duration of the program, and the expected outcomes.

Other key strategies can promote the mental health of people with diabetes more practically. The education of patients by health professionals should not have the sterile enhancement of knowledge as its sole objective. Instead, it should try to facilitate patients’ acceptance of the new circumstances of their lives. Initially, the effort should be focused on minimizing the distress caused by living with the disease. This can be achieved by fragmenting everyday tasks into manageable, small, and discrete parts and setting priorities with special attention to essential issues. Later on, the improvement of skills in self-care with ongoing information and assessment, in collaboration with clinicians, can further optimize stress-coping abilities. The provision of support by the state with resources and logistics is particularly significant in this effort.

Diabetes is a complex condition that requires actions by a multidisciplinary team of clinicians. The role of nurses is a key component in this endeavor. They should have leading duties in educating and supporting patients, as well as in interconnecting somatic and emotional care services. It is therefore appropriate to have nurse-led clinical teams that can act as case managers. Moreover, nurses should actively participate in research and professional training. The primary responsibilities of medical practitioners, including diabetologists, general practitioners, and physicians of other specialties, are to provide treatment, promote lifestyle modifications, monitor health parameters, and prevent complications, always with a focus on the special physical and emotional needs of each patient. Integrating interprofessional collaboration is certainly necessary.

In general, patients with diabetes should primarily be able to manage the emotional aspects of their disease during routine diabetes care. Therefore, diabetes specialists must have some degree of expertise in recognizing and dealing with psychological issues related to or interacting with diabetes mellitus. However, some mental health problems require referral to psychiatric care. In these cases, integrated psychosomatic–psychotherapeutic treatment can alleviate distress and depressive symptoms and assist in regulating glycemic control [94]. If needed, pharmacological agents can be incorporated into the treatment plan to address different aspects of emotional needs. Emergency services should also be available when a crisis occurs.

In sum, a holistic approach to caring for diabetes considers a patient’s emotional needs and social circumstances in addition to selecting the appropriate anti-diabetic medication. The approach to managing the well-being of patients with diabetes is summarized in Table 2. The capability to sense a patient’s mental condition is important when adapting individualized advice and clinical targets [95,96]. Just as the management of hyperglycemia must be reviewed and adjusted frequently to achieve adequate glycemic control, finding the right mental health intervention may require time and effort.

## 7. Conclusions

Both type 1 and type 2 diabetes mellitus represent a set of demanding biological and psychosocial challenges that place patients at risk for a series of psychological conditions, including diabetes-specific emotional disturbances and mental health disorders. In a reciprocal connection, emotional distress and psychiatric disorders increase the risk for glycemic deregulation and adverse diabetic outcomes. The implications for clinical practice are enormous. In the modern context of holistic care, specialists in diabetes management ought to be well equipped with the skills needed to better understand and intervene in the interplay between mental health and diabetes. For this purpose, innovative clinical strategies are required to refine many of the traditional methods of care. Nowadays, treating patients with diabetes extends far beyond the administration of suitable anti-hyperglycemic agents and didactic education on lifestyle. Therapeutic plans should also offer patients solutions to cope with the emotional aspects of diabetes.

## Figures and Tables

**Figure 1 healthcare-12-01457-f001:**
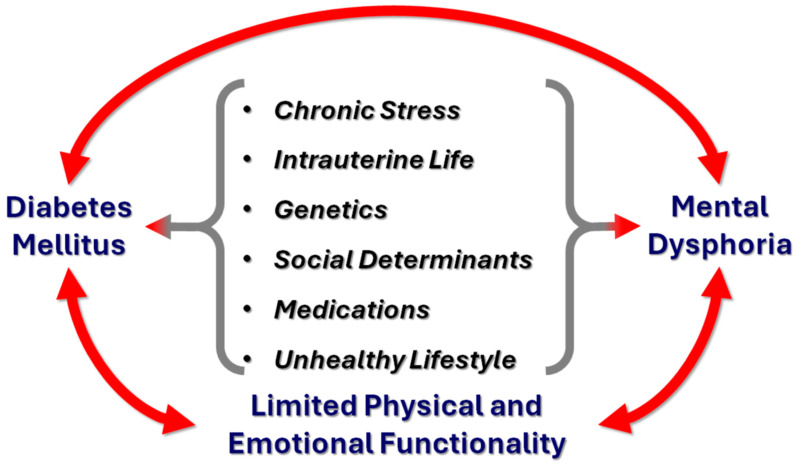
Diabetes and mental/psychological dysphoria can lead to restrictions in the physical and emotional functionality of those who suffer, forming a self-feeding cycle of interaction.

**Figure 2 healthcare-12-01457-f002:**
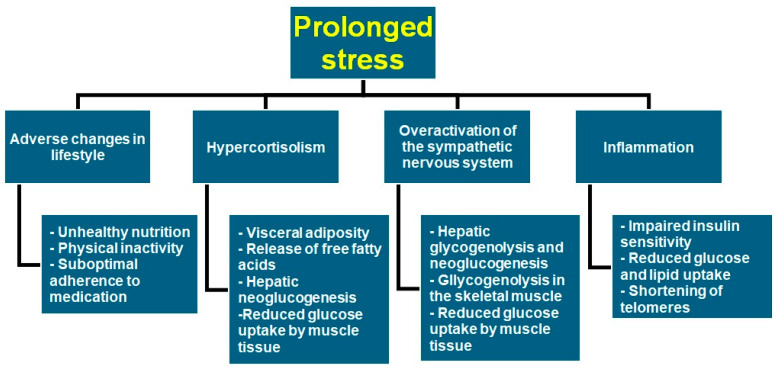
The effect of stress-related physiological mechanisms on glycemia.

**Figure 3 healthcare-12-01457-f003:**
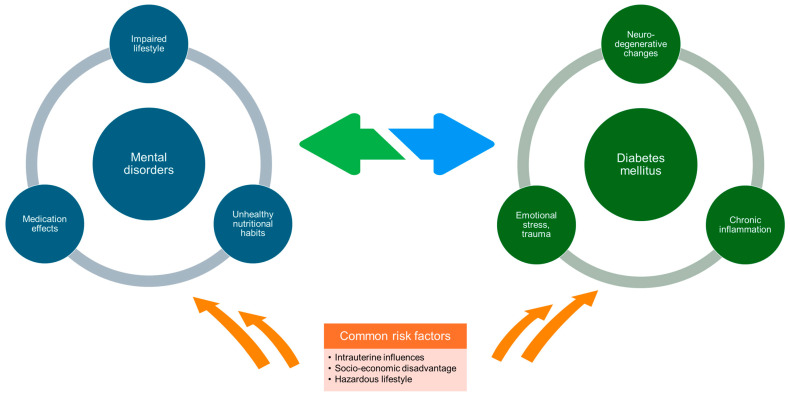
Pathways of interaction between diabetes mellitus and mental disorders.

**Table 1 healthcare-12-01457-t001:** The results of the electronic search.

Type of Study	References
Opinion piece	[30,33,34,40,45,49,64,96]
Case report/study	[72,79]
Primary research	[21,23,24,25,26,27,37,38,47,48,53,65,76,82,91,95]
Guidelines	[17,19,92]
Review	[18,20,22,28,29,31,32,35,36,39,41,42,43,44,46,52,55,56,57,58,59,61,62,63,66,67,68,69,70,71,73,75,78,80,81,83,86,87,89,90,93]
Meta-analysis	[50,51,54,60,73,75,84,85,88,94]

We have to note that some of the articles fall into two categories. Specifically, both the opinion pieces and the guidelines were largely based on literature reviews, whereas the meta-analyses included a prior systematic review.

**Table 2 healthcare-12-01457-t002:** Approach to managing the well-being of patients with diabetes.

Psychosocial Support	Guidance on Self-Care	Educates on blood glucose monitoring, diet, physical activity, and treatment adherence Provides individualized instructions to enhance self-care and emotional adjustment
	Cognitive Behavioral Therapy	Reorganizes dysfunctional thoughts and changes behaviors Focuses on optimism and resilience Improves self-care, mood, and quality of life
	Social Supportive Networks	Offers emotional, material, and practical support Formal (insurance, social services) and informal (family, friends) networks improve outcomes, especially for the vulnerable
Psychopharmacological Agents		Antidepressants Antipsychotics Anxiolytics
Recommendations for Clinical Practice		Holistic care includes emotional needs and social circumstances Diabetes specialists should recognize and manage psychological issues, with psychiatric referrals when necessary Address both physical and mental health in diabetes management Psychoeducation: disease information, acquirement of problem-solving and communication skills, increase in self-confidence Education should help patients accept and manage their condition State support is essential

## Data Availability

Not applicable.

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
