# Peer review of "The Interrelationship between Diabetes Mellitus and Emotional Well-Being: Current Concepts and Future Prospects"

_healthcare, 2024, doi:10.3390/healthcare12141457_

Round 1

Reviewer 1 Report

Comments and Suggestions for Authors

Diabetes mellitus is a lifelong metabolic disorder that impacts people's well-being and biopsychosocial status. Psychiatric problems and diabetes mellitus have a complex, reciprocal interaction in which one condition affects the other. In this narrative review, the authors provide an overview of the literature on the psychological effects of diabetes, explain the evaluation of emotional disorders in the setting of diabetes, and suggest interventions aimed at enhancing both mental and physical health.

Biomechanism figures for diabetes mellitus which can cause psychiatric problems must be provided to make it easier to understand this manuscript.

There needs to be a table that presents the latest research data that diabetes can cause psychiatric problems.

Author Response

Reviewer #1

Diabetes mellitus is a lifelong metabolic disorder that impacts people's well-being and biopsychosocial status. Psychiatric problems and diabetes mellitus have a complex, reciprocal interaction in which one condition affects the other. In this narrative review, the authors provide an overview of the literature on the psychological effects of diabetes, explain the evaluation of emotional disorders in the setting of diabetes, and suggest interventions aimed at enhancing both mental and physical health.

The authors would like to thank the reviewer for his/her contribution to the evaluation of the article.

Biomechanism figures for diabetes mellitus which can cause psychiatric problems must be provided to make it easier to understand this manuscript.

According to the reviewer’s suggestion, two new figures have been added to the revised text. Hopefully, these additions can illustrate the complex biological mechanisms that connect diabetes mellitus and emotional disorders.

There needs to be a table that presents the latest research data that diabetes can cause psychiatric problems.

A relevant table, with the retrieved research which was used in this work, has been added to the revised paper.

Reviewer 2 Report

Comments and Suggestions for Authors

This is a well-written, detailed and comprehensive narrative review; table 1 concludes the paper as a useful summary. I have a few comments regarding the comprehensiveness of the presentation of the methodology and the structure of the article:

Abstract, lines 24, 25: I suggest to write "a comprehensive biopsychosocial approach should be taken, and where appropriate, psychopharmacological therapies or psychotherapy should be applied." This would underline the need to incorporate the framework/measures mentioned into clinical practice.

p. 2, "1.2 methods": I would recommend to use a higher numbering for the methods part, i.e., 2, since the introduction and methods sections should be clearly separated.

p. 3, lines 83 et seq.: It would be helpful to have some more details on how the review was conducted such as: What time period was searched, which types of articles were included, how many articles were ultimately included? Perhaps the complete list of search terms used in the literature search might be provided as a supplement.

p. 3, line 102: Shouldn't it be "...manifests IN..."? Please check.

p. 3, section 2: I wonder if this section, or at least some parts of the information given here, would not be better integrated in the introduction section and presented there, as the authors intend to focus on the emotional and psychosocial consequences and correlates of diabetes.

p. 5. line 220: It would be helpful if the authors included a reference for the statement that there is an increased prevalance of eating disorders in diabetes patients, even though this comorbidity is addressed in more detail in the following section.

p. 6, lines 250 et seq.: Is the information presented in this entire paragraph based solely on references 52-54? If not, please add further appropriate references.

p. 6, line 270: I would suggest to write "Anxiety disorders usually involve..." instead of "Anxiety disorder usually involves...".

p. 8, section 5.1.1: It would be useful to mention specific, structured interventions/settings to promote disease/self-management such as medical rehabilitation, patient education or disease management programs, which are common in various healthcare systems. The authors mention psychoeducation in the recommendations for clinical practice, but should consider outlining it here as well.

Author Response

Reviewer #2

This is a well-written, detailed and comprehensive narrative review; table 1 concludes the paper as a useful summary. I have a few comments regarding the comprehensiveness of the presentation of the methodology and the structure of the article:

The authors would like to thank the reviewer for his/her evaluation.

Abstract, lines 24, 25: I suggest to write “a comprehensive biopsychosocial approach should be taken, and where appropriate, psychopharmacological therapies or psychotherapy should be applied.” This would underline the need to incorporate the framework/measures mentioned into clinical practice.

These recommendations are totally correct and the revised text has been changed accordingly.

“1.2 methods”: I would recommend to use a higher numbering for the methods part, i.e., 2, since the introduction and methods sections should be clearly separated.

The numbering and the order of sections have been appropriately changed in order to reflect a better structure of the text.

lines 83 et seq.: It would be helpful to have some more details on how the review was conducted such as: What time period was searched, which types of articles were included, how many articles were ultimately included? Perhaps the complete list of search terms used in the literature search might be provided as a supplement.

The specific part of the text has been revised to include details of the literature search. In addition, a table containing the retrieved papers has been added to the revised paper.

line 102: Shouldn't it be “...manifests IN...”? Please check.

The wording was found to be correct at this particular point.

section 2: I wonder if this section, or at least some parts of the information given here, would not be better integrated in the introduction section and presented there, as the authors intend to focus on the emotional and psychosocial consequences and correlates of diabetes.

This is certainly a correct suggestion. The order of sections has been changed to better present the introductory information.

line 220: It would be helpful if the authors included a reference for the statement that there is an increased prevalence of eating disorders in diabetes patients, even though this comorbidity is addressed in more detail in the following section.

Further references have been added in order to document the high prevalence of eating disorders among people with diabetes.

lines 250 et seq.: Is the information presented in this entire paragraph based solely on references 52-54? If not, please add further appropriate references.

The references 52,53,54 refer to the entire specific patagraph of our article.

line 270: I would suggest to write "Anxiety disorders usually involve..." instead of "Anxiety disorder usually involves...".

We totally agree with this recommendation; the text has been accordingly changed.

section 5.1.1: It would be useful to mention specific, structured interventions/settings to promote disease/self-management such as medical rehabilitation, patient education or disease management programs, which are common in various healthcare systems. The authors mention psychoeducation in the recommendations for clinical practice, but should consider outlining it here as well.

New information has been added to the revised text, as the reviewer suggested.

Reviewer 3 Report

Comments and Suggestions for Authors

Dear authors,

Thanks for the effort.

Here are some general comments:

This is indeed an important topic for the individiuals with DM. I would have expected potantial mechanisms behind the mental disorders in to to what has been discussed in the paper. 

Also, it would be great to add a compehensive figure which explains the reasons of the mental healht problems: could be both related to diaase itself or oathological mechanisms (like you discussed in depression section) - recommendation/approaches to be made. 

Specific comments: 

- Proofreading - some sentences are too simple and there are some grammar errors. Would be good to get proof done. 

- Introduction is divided in different section, however the first paragraph looks too short, should be expanded. 

- Method: 

 "The relevant literature  was searched in the Medline and Google Scholar databases with the combined use of a series of terms, including “diabetes mellitus”, “distress”, “mental health”, “quality of life”,  and “psychological interventions”. Articles that served the narrative of this review were then purposefully selected."

please explain why did not add depression etc to keywords? should have been added. 

- did you discard pre-clinical studies?

- please icnldued dates for your search - both the date you conduscted this search and the date you filtered on Medline etc. 

- please explain how did to exclude / include a paper? 

- in general: method should be improved. 

Recommendations for clinical practice: It should be well defined what are the roles of MDs, nurses, etc. "health professionals" alone is not clear. 

Author Response

Reviewer #3

This is indeed an important topic for the individuals with DM. I would have expected potential mechanisms behind the mental disorders into what has been discussed in the paper.

The authors thank the reviewers for his/her views of the article.

Also, it would be great to add a comprehensive figure which explains the reasons of the mental health problems: could be both related to disease itself or pathological mechanisms (like you discussed in depression section) - recommendation/approaches to be made.

These are indeed significant issues regarding the content of the article. Two figures have been inserted in the revised text to outline the correlation between mental disorders, diabetes, and associated pathological mechanisms. In addition, the section of recommendations has been expanded to include critical information.

Specific comments:

Proofreading - some sentences are too simple and there are some grammar errors. Would be good to get proof done.

The revised text has been checked for grammatical errors.

Introduction is divided in different section, however the first paragraph looks too short, should be expanded.

The first paragraph has been expanded and the order of sections has been changed.

Method:

“The relevant literature was searched in the Medline and Google Scholar databases with the combined use of a series of terms, including “diabetes mellitus”, “distress”, “mental health”, “quality of life”,  and “psychological interventions”. Articles that served the narrative of this review were then purposefully selected.” please explain why did not add depression etc to keywords? should have been added.

This comment is to the point. Unfortunately, the specific keyword was not used as a separate search term. However, numerous relevant titles emerged about the correlation of diabetes with depression. These papers were considered to be adequate for covering the paper’s inquiry and consequently, an additional search was not conducted.

Did you discard pre-clinical studies?

Pre-clinical studies were not discarded intentionally. They were not among the papers that served the narrative of this review.

Please include dates for your search - both the date you conducted this search and the date you filtered on Medline etc.

The relevant dates have been added in the new form of the text.

Please explain how did to exclude / include a paper?

Clarifications have been added to the text for this issue.

In general: method should be improved.

This part of the text has been extended to provide a better piece of information.

Recommendations for clinical practice: It should be well defined what are the roles of MDs, nurses, etc. "health professionals" alone is not clear.

An additional paragraph has been inserted in the section of recommendations that elaborates on the role of nurses and medical doctors..

Round 2

Reviewer 2 Report

Comments and Suggestions for Authors

Dear authors,

thank you for revising your manuscript. My comments and suggestions have been adequately taken into account. Please consider the following minor points:

p. 4, line 133: Please provide the actual number of hits in the literature search (instead of "more than 5000 titles").

p. 4, line 136: Please delete the hyphen in "me-ta-analyses".

Regarding my comment on references on the prevalence of eating disorders in diabetes patients: Are the references [47, 48] on p. 7, line 247 those that were added in the revision (they are not marked in red)?

Author Response

[1]. My comments and suggestions have been adequately taken into account. Please consider the following minor points:

Once again, we would like to thank the reviewer for his/her evaluation of our article. We really appreciate his/her opinion.

[2]. p. 4, line 133: Please provide the actual number of hits in the literature search (instead of "more than 5000 titles").

The actual number is provided in the revised text.

[3]. p. 4, line 136: Please delete the hyphen in "me-ta-analyses".

This typo (along with some other ones) was corrected.

[4]. Regarding my comment on references on the prevalence of eating disorders in diabetes patients: Are the references [47, 48] on p. 7, line 247 those that were added in the revision (they are not marked in red)?

We are sorry for the omission. Indeed, references 47 and 48 are the additional references and they should have been marked in red. This has been rectified in the revised version of our manuscript.

Reviewer 3 Report

Comments and Suggestions for Authors

Dear authors,

Thanks a lot for addressing the comments. I do not have further comments on the paper.

Author Response

[1]. Thanks a lot for addressing the comments. I do not have further comments on the paper.

Once again, we would like to thank the reviewer for his/her evaluation of our article. We really appreciate his/her opinion.